# Secondary Metabolites and Their Cytotoxic Activity of *Artemisia nitrosa* Weber. and *Artemisia marschalliana* Spreng.

**DOI:** 10.3390/molecules27228074

**Published:** 2022-11-21

**Authors:** Aizhan Kazymbetova, Magzhan Amangeldi, Aliya Nurlybekova, Ulpan Amzeyeva, Kunbike Baktybala, Chun-Ping Tang, Chang-Qiang Ke, Sheng Yao, Yang Ye, Janar Jenis

**Affiliations:** 1The Research Center for Medicinal Plants, Al-Farabi Kazakh National University, al-Farabi ave. 71, Almaty 050040, Kazakhstan; 2University of Chinese Academy of Sciences, Beijing 100049, China; 3State Key Laboratory of Drug Research, Shanghai Institute of Materia Medica, Chinese Academy of Sciences, Shanghai 201203, China; 4Research Institute for Natural Products & Technology, Almaty 050046, Kazakhstan

**Keywords:** *Artemisia nitrosa*, *Artemisia marschalliana*, sesquiterpene lactone, coumarin, flavonoid, phenolic compound, cytotoxicity, HL-60, A-549

## Abstract

As a promising source of biologically active substances, the *Artemisia* species from Kazakhstan have not been investigated efficiently. Considering the rich history, medicinal values, and availability of the *Artemisia* plants, systematic investigations of two *Artemisia* species growing in the East Kazakhstan region were conducted. In this study, one new germacrane-type sesquiterpene lactone (**11**), together with 10 known sesquiterpenes and its dimer, were characterized from *A. nitrosa* Weber. Additionally, one new chromene derivative (**1**’) with another 12 known compounds, including coumarins, sesquiterpene diketones, phenyl propanoids, polyacetylenics, dihydroxycinnamic acid derivatives, fatty acids, naphthalene derivatives, flavones, and caffeic acid derivatives were isolated from *A. marschalliana* Spreng. All compounds were isolated and identified for the first time from these two *Artemisia* species. The structures of new compounds (**11**, **1**’) were established by using UV, TOFMS, LC–MS, 1D and 2D NMR spectroscopic analyses. The cytotoxicity of all isolated compounds was evaluated. As a result, all compounds did not show significant inhibition against HL-60 and A-549 cell lines. The sesquiterpenoids isolated from *A. nitrosa* were tested for their inhibitory activity against the LPS-induced NO release from the RAW624.7 cells, and neither of them exhibited significant activity.

## 1. Introduction

The *Artemisia* species are perennial high-vascular plants and have been used for centuries in traditional medicine [1]. The medicinal benefits of the Artemisia genus include normalizing the work of the gastrointestinal tract, especially in gastritis with low acidity, increasing appetite [2,3,4], and treating bronchial asthma [5], rheumatism [6], dermatitis [7,8], malaria [9], etc. Moreover, a few scientific publications reported that some natural *Artemisia* drugs showed promising potential to cure diseases, such as AIDS, cancer, cardiovascular diseases, and renal disorders [10,11,12]. Extensive research has resulted in the isolation of a number of bioactive secondary metabolites, such as essential oils, flavonoids, terpenes, esters, and phenolic [13,14]. Many compounds from the genus showed antimalarial, antiviral, anticancer, antipyretic, antihemorrhagic, anticoagulant, antianginal, antioxidant, antiulcer, and antispasmodic properties [1,15,16,17].

*Artemisia* is one of the largest genera in the Asteraceae family, encompassing more than 400 species, and is widely distributed all over the world [18,19]. The most significant number of species are found in Russia and China and, in Kazakhstan, 81 species were documented, with 19 being endemic, and 34 growing in the territory of Central Kazakhstan [20,21]. As is well known, plants of the *Artemisia* species have a history of manufacturing potentially cytotoxic substances. For instance, the Central Asian oncology clinics use a sesquiterpene lactone named arglabin, derived from the *A. glabella* plant growing in Central Kazakhstan, to treat various cancers [22]. ]. The pharmacologically active flavone eupatrilin, which was isolated from *A. asiatica*, has cytotoxic and chemopreventive properties [23]. Our previous work on the endemic *A. *heptaptamica** in the Almaty region of Kazakhstan revealed 13 sesquiterpene lactones, most of which showed potent inhibition against the activation of NF-kB induced by LPS [24].

Another recent study of our group has revealed that methanolic extracts of a total of nine *Artemisia* species from Central Asia showed a high potential for α-glucosidase, PTP1B, antioxidant, and BNA inhibition, which are associated with diabetes, obesity, and bacterial infections. Of these, both *A. nitrosa* and *A. marschalliana* exhibited a PTP1B inhibition around 75% at a concentration of 50 µg/mL. Similarly, both *Artemisia* species also showed the highest activities (>85%) against BNA even at lower concentration of 20 µg/mL [21]. 

*Artemisia nitrosa* Weber is native to saline desert-steppe landscapes of Kazakhstan, southern Siberia, and Mongolia, with secondary distribution in Transbaikalia [25]. However, *A. nitrosa* is a poorly studied plant. Secondary metabolites of *A. nitrosa*, such as sesquiterpene lactones and dimers, are being isolated and identified for the first time by our research team. 

*Artemisia marshalliana* Spreng is found in steppe meadows, steppes, and pine forests throughout the Far East, Siberia, the Caucasus, and Kazakhstan [20]. It is an Iranian traditional medicinal plant whose extracts showed antibacterial and anticancer properties in human gastric carcinoma (AGS) and L929 cell lines, while the essential oil has antimalarial properties [26,27].

This study aimed to phytochemically investigate non-explored *Artemisia* species in Kazakhstan, and resulted in the characterization of germacrene-type sesquiterpene lactones from *A. nitrosa* and phenolic compounds from *A. marschalliana*, including a total of 2 new and 23 known compounds for the first time. Their structures have been established using extensive analyses of UV, MS, 1D, and 2D NMR spectroscopic data. All compounds were evaluated for cytotoxicity against human cancer cell lines HL-60 and A-549. 

## 2. Materials and Methods

### 2.1. General Experimental Procedures

To distinguish a certain substance, the combination of NMR (1D and 2D) analytical techniques with other experimental methods, such as LC–MS, UV, IR, preparative HPLC, and semi-preparative HPLC were used. A Shimadzu UV-2550 UV–vis spectrophotometer is used for the measurement of UV spectra. The IR spectra are registered on a Thermo Nicolet FTIR IS 5 spectrophotometer. The HR-ESIMS spectra were measured on a Waters Synapt G2-Si Q-TOF instrument with a Waters BEH C18 column (1.7 μm, 2.1 mm× 50 mm, CH_3_CN:H_2_O with 0.1% formic acid, from 5% to 95%, 0–9 min, flow rate 0.4 mL/min, 45 °C). Analytical HPLC was performed on a Waters e2695 system equipped with a Waters 2998 photodiode array detector (PDA), a Waters 2424 evaporative light-scattering detector (ELSD), and a Waters 3100 MS detector, using a Waters Sunfire RP C18 column (5 μm, 4.6 mm × 150 mm, CH_3_CN:H_2_O with 0.1% formic acid, from 5% to 95%, 0–25 min, flow rate 1.0 mL/min, 30 °C). Preparative HPLC was run on a Waters system equipped with a Waters 2767 autosampler, a Waters 2545 pump, a Waters 2489 PDA and an Acuity ELSD using a Waters Sunfire RP C18 column (5 μm, 30 mm × 150 mm, flow rate 30 mL/min).

The NMR spectra were recorded on a Bruker Avance III (Bruker, Zurich, Switzerland) using a 500 M NMR spectrometer with TMS as the internal standard. The chemical shift (δ) values were given in ppm and coupling constants (J) in Hz. All solvents used for CC were of at least analytical grade (Shanghai Chemical Reagents Co., Ltd., Shanghai, China), and solvents used for HPLC were of HPLC grade (Merck KGaA, Darmstadt, Germany).

Column chromatography (CC) was performed on MCI gel CHP20P (75−150 μm, Mitsubishi Chemical Industries, Tokyo, Japan), Econosep C18 60A (50 μm, DIKMA, Beijing, China), Sephadex LH-20 (Pharmacia Biotech AB, Uppsala, Sweden), and silica gel (100−200 and 300−400 mesh, Qingdao Haiyang Chemical Co., Ltd., Qingdao, China). The TLC was carried out on precoated silica gel 60 F254 aluminum sheets (Merck, Darmstadt, Germany), and the TLC spots were viewed at 254 nm and visualized using 5% sulfuric acid in alcohol containing 10 mg/mL of vanillin. 

### 2.2. Plant Materials

Here, *A. nitrosa* and *A. marschalliana* were gathered from East Kazakhstan at the end of July 2020 and identified by experts of the Republican State Enterprise on the subject of economic management at the “Institute of Botany and Phytointroduction” of the Committee of Forestry and Wildlife of the Ministry of Ecology, Geology, and Natural Resources of the Republic of Kazakhstan. A sample of *A. nitrosa* (No. ANI-07) and a sample of *A*. *marschalliana* (AMA-07) were deposited in the herbarium of the Research Center for Medicinal Plants, Faculty of Chemistry and Chemical Technology, Al-Farabi Kazakh National University, Almaty, Kazakhstan (Appendix A). The air-dried whole plants of *A. nitrosa* (14 kg) and *A. marschalliana* (13 Kg) were cut into small pieces and stored at room temperature.

### 2.3. Extraction and Isolation of A. nitrosa

The air-dried whole plants of *A. nitrosa* (14 kg) were ground and extracted by 95% aqueous EtOH at room temperature three times (three days each). The EtOH extract was dried with evaporation under reduced pressure. The residue was suspended in H_2_O and partitioned with petroleum ether (PE), CHCl_3_, and EtOAc (EA), successively (Figure 1). The CHCl_3_ fraction (355 g) was subjected to an AB-8 column chromatography (CC), eluted with aqueous EtOH in a gradient manner (30, 50, 70, 80, and 95%), obtaining fractions 1–6. Fraction 2 of 50% of EtOH (110 g) was applied to a polyamide column using H_2_O, 30, 50, 70, 95% of EtOH/H_2_O (*v*/*v*) as an eluent. The water fraction (56 g) was further fractionated over a Sephadex LH-20 column eluted with CHCl_3_–MeOH (1:1) to give five fractions (A1–A5). Subfraction A4 was subjected to CC on a ODS medium pressure column eluted with MeOH (through a gradient from 35 to 100% aqueous MeOH) to obtain A4A–A4H + J subfractions. The subfraction A4E (3.9 g) was passed through the silica gel column using a solvent mixture of CHCl_3_:MeOH (from 100:1 to 1:1) to give subfractions A4E1–A4E10. Further purification of subtraction A4E was performed by repeated CC over silica gel, giving seven pure compounds, as follows: **1** (110 mg), **2** (54 mg), **3** (23 mg), **4** (17 mg), **5** (38 mg), **6** (7 mg), and **7** (8 mg). Subfraction A4F (602 mg) was purified first by CC over silica gel, and then preparative HPLC (with a gradient of CH_3_CN/H_2_O) to obtain compounds **8** (3 mg) and **11** (2 mg). In a similar way, subfraction A4C (667 mg) was treated and finally purified by semi-preparative HPLC using CH_3_CN/H_2_O (15 min, from 50% to 75%, flow rate 3.0 mL/min) to obtain compound **12** (4 mg). Subfraction A4D (3.2 g) was applied to a silica gel column eluted with different solvent systems (CHCl_3_:acetone, and CHCl_3_:MeOH in a ratio of 100:1 to 10:1) to obtain compounds **9** (1 mg) and **10** (3 mg) (Figure 1). 

### 2.4. Extraction and Isolation of A. marschalliana

The air-dried whole plants of *A. marschalliana* (13 kg) were powdered and extracted by 95% aqueous EtOH at room temperature three times (3 day each). The EtOH extract was dried with evaporation under reduced pressure. The residue was suspended in water and partitioned with petroleum ether (PE), CHCl_3_, and EA, respectively (Figure 2). The obtained CHCl_3_ fraction (50 g) was applied to a silica gel column (200–300 mesh) eluted with aqueous EtOH (in a gradient manner from 20% to 95%) to obtain 12 subfractions. The subfractions 7 and 8 (3.579 g) were subjected to CC over Sephadex LH-20 using MeOH as mobile phase, giving five subfractions (7 and 8A to 7 and 8E). Fraction 4 (890 mg) was further applied to CC over silica gel (200–300 mesh) eluting with a gradient solvent system of PE/EtOAc to give compound **1**’ (11 mg). Subfraction 5 (602 mg) was passed through a column of silica gel (200–300 mesh, PE/EtOAc) to give subfractions 5A–5H. Subsequently, subfraction 5F was purified by preparative HPLC using CH_3_CN/H_2_O to yield compound **2**’ (80 mg). Fraction 10 (2.49 g) was subjected to CC over ODS using aqueous MeOH to yield 13 subfractions (Fr10A–Fr10M). Furthermore, subfraction 10J was subjected to CC over Sephadex LH-20 to obtain compound **3**’ (70 mg). The obtained PE fraction (110 g) was further extracted with 80% aqueous MeOH (34.3 g), which was applied to a polyamide gel column eluted with aqueous EtOH in a gradient manner (20%, 40%, to 95%) to obtain five subfractions (A-E). Subfraction B (3.5 g) was further applied to silica gel CC eluting through a gradient with a solvent mixture of PE/EtOAc (10:1, 1:1, 0:1) to give 10 subfractions (B1–B10). Subfraction B8 (933 mg) was passed through a column of Sephadex LH-20 (eluted with MeOH), giving four subfractions (A to D). Subfraction B8D (460 mg) was applied to CC over ODS (aqueous MeOH, from 98:2 (*v*/*v*) to 25:75 (*v*/*v*)) to obtain subfractions B8D1–B8D5. Subsequently, fraction B8D2 was purified by silica gel CC using a gradient solvent system petroleum ether/EtOAc to obtain compound **4**’ (7 mg), and then subfractions were subjected to preparative HPLC (CH_3_CN/H_2_O) to yield compounds **5**’ (5 mg), **6**’ (27 mg), **7**’ (10 mg), and **8**’ (3 mg). Subfraction C was treated first by CC over polyamide, and then applied to CC over Sephadex LH-20 eluting with MeOH, obtaining six subfractions (C1 to C6). Subfraction C6 was purified using silica gel CC (CHCl_3_/EtOAc), obtaining five fractions (C6A to C6E), and then fraction C6E was applied to an Auto-P machine to obtain compounds **9**’ (1 mg), and compound **10**’ (8 mg). Subfraction C6C was also applied on an Auto-P machine using CH_3_CN/H_2_O as an eluent to obtain compound **11**’ (3 mg) and compound **12**’ (2 mg). Fraction C5 was subjected on CC over ODS using CH_3_CN/H_2_O as an eluent to obtain compound **13**’ (2 mg) (Figure 2).

### 2.5. Cytotoxicity Assay

The cytotoxic effects *of A. nitrosa* and *A. marschalliana* were determined using the colorimetric (CCK8) method [28] and the sulforhodamine B (SRB) protein staining method [29]. The CCK8 method was used to detect the growth inhibition of HL-60 cell lines. Cells with a logarithmic growth phase were seeded into a 96-well culture plate at a specific density (90 μL per well); after culturing overnight, different concentrations of drugs were added for 72 h. Three replicate wells were set up for each concentration, which corresponds to concentrations of vehicle control and cell-free zero adjustment wells. Then, 10 μL of CCK-8 was added to each well. After incubating for 2~3 h in the incubator, the SpectraMax 190 microplate reader was used to measure the optical density (OD value) at the 450 nm wavelength.

The compound’s inhibitory effect on the proliferation of A549 cells was detected by the sulforhodamine B (SRB) protein staining method. The specific steps are as fol-lows: A549 cells in the logarithmic growth phase are seeded into a 96-well culture plate at an appropriate density, 90 μL per well; after overnight culture, different con-centrations of compounds (DMSO concentration less than 0.5%) are added for 72 h, each set has three wells for each concentration, and a solvent control group (negative control) is set. After the effect is over, the culture medium is discarded, and 10% (*w*/*v*) trichloroacetic acid (100 μL/well) is added; the solution is fixed at 4 °C for 1 h, then washed with distilled water five times, before being dried at room temperature. Then, we added 100μL of SRB solution (4 mg/mL, dissolved in 1% glacial acetic acid), incu-bated it for 15 min at room temperature, rinsed with 1% glacial acetic acid five times to wash away unbound SRB, and added 10 mM Tris solution 100 μL to each well after drying at room temperature, before using a full-wavelength microplate reader Spec-traMax 190 at the 515 nm wavelength to determine the OD value.

The inhibitory rate of the compound on cell proliferation is calculated by the following formula: Inhibition Rate = [1 − (*OD*_cpd_ − *OD*_untreated_)/(*OD*_LPS_ − *OD*_untreated_)] ∗ 100%.

### 2.6. Cell Viability Evaluation

Here, RAW264.7 cells were seeded into 96-well plates at a concentration of 1 × 10^4^ cells per well and allowed to adhere to the bottom of the plate overnight. Then, the cells were treated with different concentrations of compounds for 18 h. The cell viability was determined by MTT assay, as described previously [30]. Then, cell viability was determined by incubation with DMEM containing MTT (1 mg·mL^−1^) for 4 h, followed by dissolving the formazan crystals with 150 μL DMSO. The absorbance at 540 nm was measured by a SpectraMax M5 microplate reader (Molecular Devices, San Jose, CA, USA).

### 2.7. Measurement of Nitric Oxide (NO) Production

Here, RAW264.7 cells were seeded into 96-well plates (1 × 10^4^ cells per well) and allowed to adhere for 24 h. The cells were then treated with different concentrations of compounds or vehicles (DMSO) followed by stimulation with 1 μg·mL^−1^ lipopolysaccharide (LPS, Sigma-Aldrich, St. Louis, MO, USA). The DMSO was used as the vehicle, with the final concentration of DMSO being maintained at 0.1% of all cultures. After 18 h of incubation, the supernatant was collected to determine the NO content using the Griess reagent (Sigma-Aldrich, St. Louis, MO, USA) as described previously [31]. The absorbance at 490 nm was measured by a SpectraMax M5 microplate reader (Molecular Devices, San Jose, CA, USA).

## 3. Results and Discussion 

### 3.1. Structural Elucidation of Compounds from A. nitrosa

A new germacranolide type sesquiterpene lactone (**11**), together with 10 known sesquiterpene lactones (**1**–**10**) and 1 known dimeric sesquiterpene lactone (**12**) (Figure 3) were separated from *A. nitrosa*. After detailed spectroscopic analysis (1D, 2D NMR, LC–MS, TLC) and comparing with the literature data, the known compounds were identified as 1*β*,9*β*-Dthydroxyeudesm-3-en-5*α*,6*β*,*11β*-λ2,6-olide (**1**) [32,33], decahydro-5,6-dihydroxy-3,5*α*-dimethyl-9-methylenenaphtho[1,2-*β*]furan-2(3H)-one (**2**) [34], deacetylherbolide D (**3**) [32], deacetyl derivative of herbolide A (**4**) [32],1*β*-ydroperoxy-9*β*-acetoxygermacra-4,10(14)-dien-6*β*,11*β*-12,6-olide (**5**) [32], herbolide B (**6**) [32], 11*β*,13-dihydroridentin 3-acetate (**7**) [35], balchanolide (**8**) [36], 11,13-dihydro germacronolide (**9**) [37], deacetylherbolide A (**10**) [38], and artebarrolide (**12**) [36] (Figure 3, Appendix A).

Compound **11**, obtained as a colorless oil, had a molecular formula of C_18_H_26_O_5_ on the basis of UV, ESIMS, TOFMS, LC–MS, and NMR spectroscopic data. The TOFMS showed *m*/*z* 322.087 (Appendix A); UV (MeOH) λ_max_ (log Ɛ) 256 (2.07) (Appendix A).The ^1^H NMR spectrum data displayed signals of three methyl groups (*δ*_H_ 1.58 (d, *J* = 1.3 Hz, 3H), 1.27 (d, *J* = 6.9 Hz, 3H), and 1.11 (t, *J* = 7.6 Hz, 3H)), one exocyclic methylene group (*δ*_H_ 4.76, 3.93 each d, *J* = 10.2 Hz), and a characteristic signal of a double bond (*δ*_H_ 5.13 (dd, *J* = 10.2, 1.6 Hz, 1H)) (Table 1). The ^13^C NMR and DEPT NMR spectra indicated 18 carbon resonances, including 3 methyls (*δ*_C_ 17.48, 12.55, 8.53), 5 methylenes (*δ*_C_ 37.54, 36.81, 30.76, 27.64, 114.37), 6 methines (*δ*_C_ 80.05, 78.81, 74.36, 51.05, 41.56, 121.28), and 4 quaternary carbons (*δ*_C_ 177.43, 175.06, 153.51, 145.18) (Table 1). The data suggested that compound **11** might be a germacrane-type of sesquiterpene lactone.

A comparison of NMR data of **11** and the known compound 1*β*-hydroperoxy-9*β*-acetoxygermacra-4,10(14)-dien-6*β*,11*β*-12,6-olide (**5**), reported from *A. herba-alba* [32,33] and also obtained in this study, revealed high similarities between these two compounds, except for an extra methyl group (*δ*_H_ 1.11, m; *δ*_C_ 8.53) (Table 1) present in compound **11**. Detailed analysis of the 2D NMR data of **11** further established the structure. The ^1^H-^1^H COSY correlations of H-1/H-2/H-3, H-5/H-6/H-7/H-8/H-9, H-7/H-11/H-13, and H-17/H-18 revealed the existence of four segments as shown (Figure 4). The key HMBC correlations from H-1 to C-2 and C-3, H-5 to C-3 and C-15, H-6 to C-8, H-7 to C-13, H-8 to C-7 and C-11, H-14 to C-1, C-9, and H-18 to C-17 further constructed the planar structure of **11**, with a propionyloxy group attached to C-9 (Figure 4). The relative configuration of **11** was inferred as the same with that of the known compound **5** by the similar chemical shifts and the similar coupling constants of H-1, H-6, H-9, and H-13 between these two compounds. Therefore, the structure of **11** was fully established, and named 1*β*-hydroperoxy-9*β*-propionoxygermacra-4, 10(14)-dien-6*β*,11*β*-12,6-olide (**11**).

Compounds **1**–**10** and **12** have already been isolated and described from other Artemisia species, such as *A. herba-alba*, *A. barrelieri*, and *A. gypsacea*. Artebarrolide (**12**), which is the first dimeric germacranolide described from *A. barrelieri* [36], was discovered for the second time in this investigation.

### 3.2. Structural Elucidation of Compounds from A. marschalliana

A total of 13 compounds were isolated and identified from the whole plant of *A. marschalliana* that grows in Kazakhstan, including coumarins, sesquiterpene diketone, phenylpropanoid, benzopuran derivative, polyacetylenic compounds, fatty acids, naphthalene derivative, flavone, and caffeic acid derivative (Figure 5). By extensive spectroscopic analysis of MS, ^1^H, and ^13^C NMR data, and comparison with previously reported data, the structures of known compounds were identified as 1 new 2,2-dimethyl-8-(19-hydroxy)prenyl-6-(12-hydroxy) vinylchromene (**1**’) [39], together with 12 known compounds, namely a methyl 3-(4′-hydroxyprenyl)-7Z-coumarate (**2**’) [40], arteordoyn A (**3**’) [41,42], 6-acetyl-2,2-dimethylchroman-4-one (**4**’) [43], 5,6,7-trimethoxycoumarin (**5**’) [41], 2-isovaleroyl-4[1-hydroxyethyl]-phenol (**6**’) [44], guayulone (**7**’) [45], diprenylated-dihydroxycinnamic acid (**8**’) [46], dehydrofalcarinol (**9**’) [44], 9,12-Octadecadienoic acid (9Z,12Z)-,(2R)-2,3-dihydroxypropyl ester (**10**’) [47], palmarumycin CP 2 (**11**’) [48], 5,7-dihydroxy-6,4’-dimethoxyflavone (**12**’) [49], and propyl caffeate (**13**’) [50] (Figure 5, Appendix A).

Compound **1**’, obtained as a colorless oil, had the molecular formula of C_18_H_22_O_3_ on the basis analysis of ESIMS and ^13^C NMR data. The UV spectrum showed maximal absorptions at 239, 274, and 318 nm (Appendix A), indicative of the presence of a conjugated aromatic ring. The ^1^H NMR spectrum showed signals of three methyls (*δ*_H_ 1.72 (s, 3H), 1.36 (s, 6H)), two characteristic signals of a double bond (*δ*_H_ 5.48 (s, 1H); 6.22 (t, *J* = 13.6 Hz, 2H), and a benzol ring. The ^13^C and DEPT NMR spectra (Table 2) displayed 18 carbon resonances ascribed to 3 methyls (*δ*_C_ 27.80, 27.80, 13.37), 2 methylenes (*δ*_C_ 27.29, 68.33), 7 methines (*δ*_C_ 146.10, 130.60, 129.44, 128.41, 124.09, 123.08, 121.53, 113.92), and 6 quaternary carbons (*δ*_C_ 171.21, 152.67, 135.27, 120.60, 125.98, 76.66) (Table 2). The HMBC spectrum revealed H-2 (5.57 (d, *J* = 9.8 Hz, 1H)) correlated to C-1, C-17, C-18; H-3 (6.22 (t, *J* = 13.6 Hz, 2H)) to C-8, C-9; correlation in benzoyl ring; moreover H-4 (7.10 (s, 1H)) to C-10; H-6 (6.96 (s, 1H)) to C-10; H-13 (5.48 (s, 1H)) to C-12 and C-14 (Figure 6).

The ^1^H-^1^H COSY correlations revealed the relations between H-2 to H-3, H-10 to H-11 and H-12 to H-13 (Figure 6). A comparison of compound **1**’ with the known compound 2, 2-dimethyl-8-prenyl-6-vinylchromene showed the presence of two hydroxyl groups located at C-12 and C-19 (*δ*_H_ 6.22, t; *δ*_C_ 114; *δ*_H_ 3.98, s; *δ*_C_ 68.22) [39]. Accordingly, the full structure of **1**’ was proposed and named 2,2-dimethyl-8-(19-hydroxy)prenyl-6-(12-hydroxy) vinylchromene.

Earlier phytochemical studies on *A. marschalliana* harvested in the Iranian prov-ince of East Azerbaijan led to the isolation and identification of a high concentration of oxygenated sesquiterpenes [26,27], which is surprising due to fewer plants growing in Kazakhstan containing sesquiterpenoid compounds.

### 3.3. Cytotoxicity Activity

The separated compounds of *A. nitrosa* and *A. marschalliana* were examined for their cytotoxicity against human myeloid leukemia HL-60 cells and A-549 human lung cancer cell lines by the CCK8 and the sulforhodamine B (SRB) protein staining methods, respectively. The results (Table 3 and Table 4) showed that monomeric sesquiterpene lactones from *A. nitrosa* showed weak cytotoxic activities against both A-549 and HL-60 cell lines, while the compounds from *A. marschalliana* did not show any effect on the growth of A-549 and HL-60 cell lines (Table 3 and Table 4).

### 3.4. Anti-Inflammatory Activity

The sesquiterpenoids isolated from *A. nitrosa* were tested for their inhibitory effects against NO production on LPS-stimulated RAW264.7 macrophages. Firstly, the cytotoxicity of compounds **1**–**11** was evaluated using the MTT assay to determine the toxicity. Most compounds did not show obvious cytotoxicity towards RAW264.7 cells up to 10 μM (Appendix A). Among the isolates, compounds **2**, **9**, and **11** showed weak NO inhibitory effects at a concentration of 2.5 μM (Appendix A). Dexamethasone (Dex) was used as the positive control.

## 4. Conclusions

In this work, a phytochemical study of the whole plants of *A. nitrosa* and *A. marschalliana* growing in Kazakhstan was carried out for the first time. Twelve compounds were purified from *A. nitrosa*, including eight germacranolides, two eudesmanolides, one guaianolide, and one sesquiterpene dimer. Among them, compound 11 is a new germacrene-type sesquiterpene lactone. Moreover, a total of 13 compounds were isolated and identified from *A. marschalliana*, including 1 new chromene derivative (1’), and other known coumarins, sesquiterpene diketone, phenyl propanoid, polyacetylene compounds, fatty acids, naphthalene derivative, flavone, and caffeic acid derivative, respectively. The results revealed the chemical constituents of these two *Artemisia* plants of Kazakhstan for the first time. Their chemical constituents differed a lot from each other. The characteristic sesquiterpenoids were disclosed from *A. nitrosa*, while *A. marschalliana* was rich in other types of structures rather than sesquiterpenoids. It should be pointed out that the previous investigation of *A. marschalliana* led to the isolation of rich content of oxygenated sesquiterpenes, which suggested a more in-depth investigation for this species. All the known sesquiterpenes (**1**–**10**, **12**) have been already reported from the *Artemisia* species, such as *A. herba-alba*, *A. barrelieri*, and *A. gypsacea*. Artebarrolide (**12**) is the first dimeric germacranolide reported from *A. barrelieri*, and it was found for the second time in this study. The biological assay of these compounds is rare in previous investigations.

In this study, the cytotoxicity assay of all isolated compounds and the anti-inflammatory assay of the sesquiterpenoids were performed. The results of the cytotoxicity assay showed that none of these compounds showed significant inhibition against A-549 and HL-60 cell lines. The sesquiterpenoids isolated from *A. nitrosa* did not show significant inhibition on the LPS-induced NO release from RAW-264.7 cells at the concentrations of 10 and 2.5 μM, which closely correlates to the anti-inflammatory activity. Compared with the compounds isolated from *A. heptapotamica* in the previous study [24], we found that the sesquiterpenoids obtained from *A. nitrosa* lack the α,β-unsaturated ketone moiety in their structures, which might be pivotal to the anti-inflammatory activity. It is obvious that more in-depth investigations are needed to discover bioactive compounds from the *Artemisia* species in Kazakhstan.

## Figures and Tables

**Figure 1 molecules-27-08074-f001:**
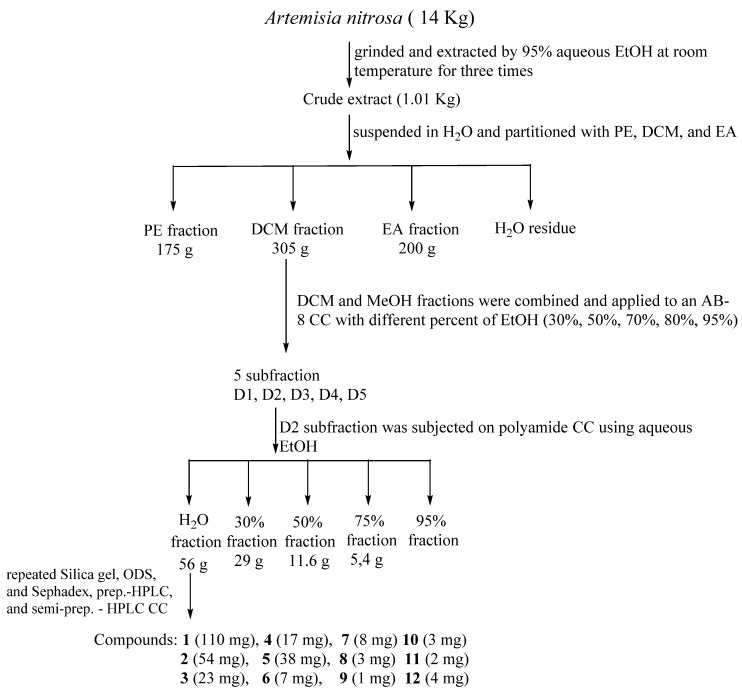
Isolation scheme of *A. nitrosa*.

**Figure 2 molecules-27-08074-f002:**
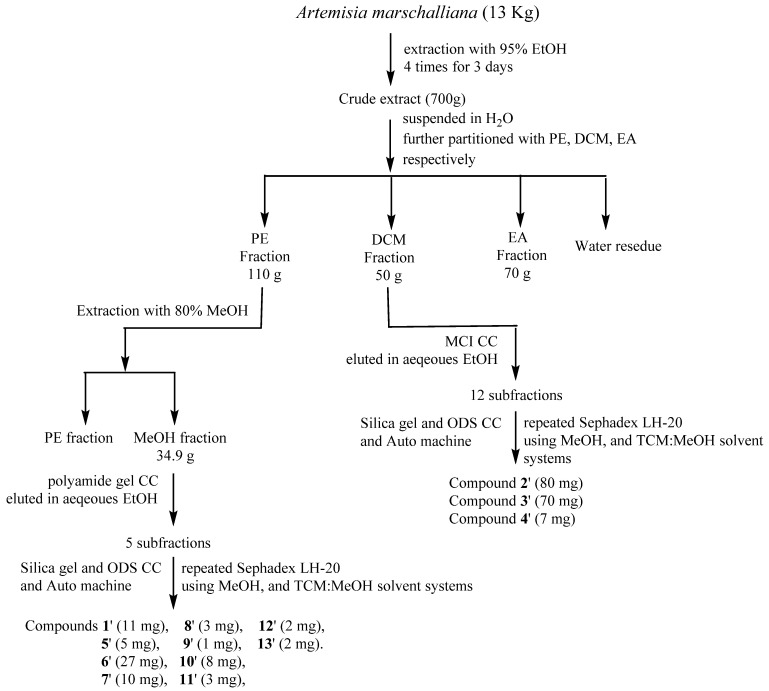
Isolation scheme of *A. marshalliana*.

**Figure 3 molecules-27-08074-f003:**
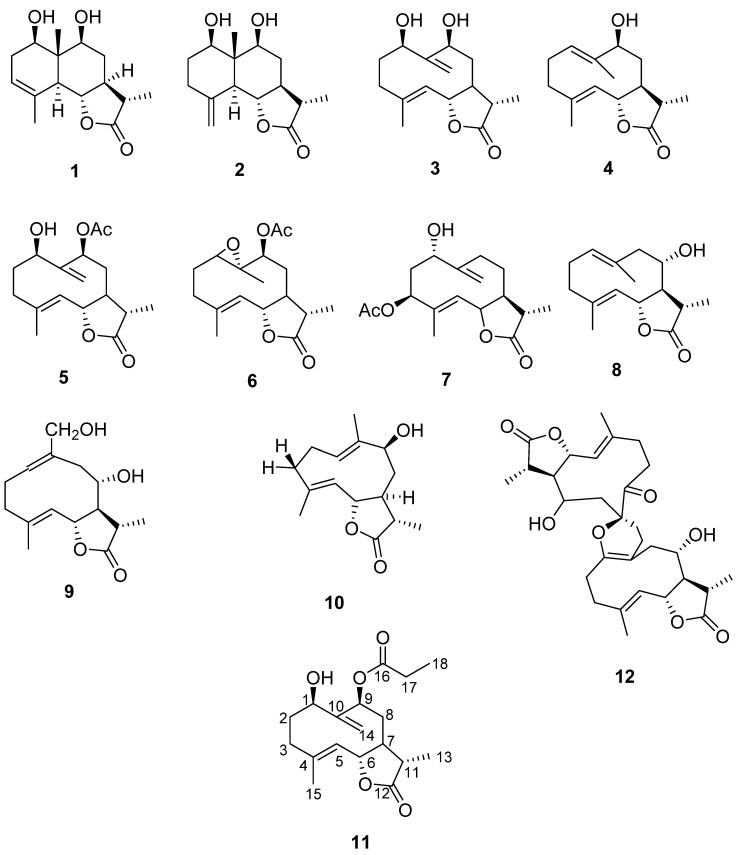
Chemical structures of compounds **1**–**12** from *A. nitrosa*.

**Figure 4 molecules-27-08074-f004:**
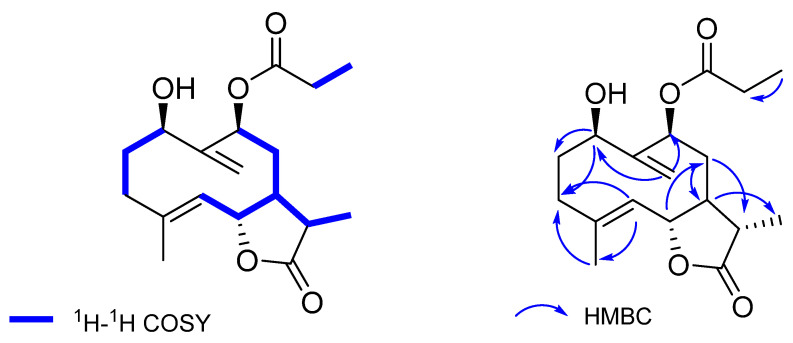
^1^H-^1^H COSY and key HMBC correlations (H→C) of compound **11**.

**Figure 5 molecules-27-08074-f005:**
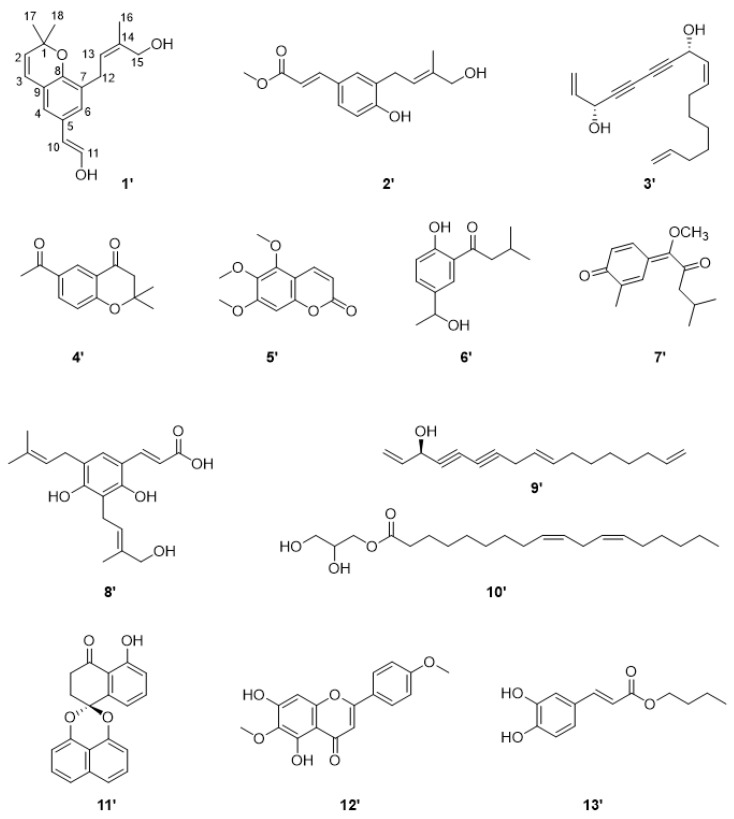
Chemical structures of compounds **1**’–**13**’ of *A. marschalliana*.

**Figure 6 molecules-27-08074-f006:**
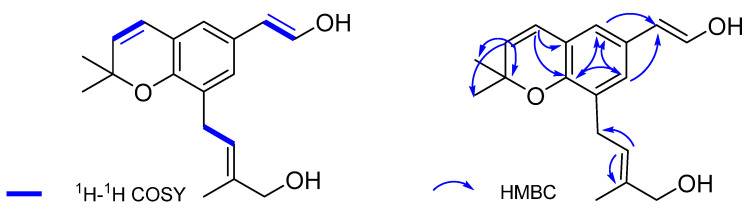
^1^H-^1^H COSY and key HMBC correlations (H→C) of compound **1**’.

**Table 1 molecules-27-08074-t001:** NMR Data for **11** (500 MHz for ^1^H and 125 MHz for ^13^C, in CDCl_3_, *δ* in ppm, *J* in Hz).

Positions	^1^H	^13^C *
1	3.93 dt (10.5, 1.4)	74.8
2	2.18–2.04 m	31.34
3	2.36–2.21 m	37.94
4	-	153.51
5	5.13 dd (10.2, 1.6)	121.28
6	4.34 t (9.8)	80.05
7	2.36–2.21 m	41.56
8	1.97–1.80 m	36.81
9	4.76 dt (10.2, 1.5)	78.81
10	-	145.18
11	1.97–1.80 m	51.05
12	-	175.06
13	1.27 d (6.9)	12.55
14	5.42 d (1.2); 5.34 br s	114.37
15	1.58 d (1.3)	17.48
16	-	177.43
17	2.36–2.21 m	27.64
18	1.11 t (7.6)	8.53

* The assignments were based on HSQC and HMBC data.

**Table 2 molecules-27-08074-t002:** NMR Data for **1**’ **(**500 MHz for ^1^H and 125 MHz for ^13^C, in CDCl_3_, *δ* in ppm, *J* in Hz).

Positions	^1^H	^13^C *
1	-	76.63
2	5.57 (d, *J* = 9.8 Hz, 1H)	130.60
3	6.22 (t, *J* = 13.6 Hz, 2H)	121.53
4	7.10 (s, 1H)	129.44
5	-	128.41
6	6.96 (s, 1H)	123.08
7	-	135.27
8	-	152.67
9	-	120.60
10	6.22 (t, *J* = 13.6 Hz, 2H)	113.92
11	7.58 (d, *J* = 15.6 Hz, 1H)	146.23
12	3.25 (d, *J* = 7.2 Hz, 2H)	27.29
13	5.48 (s, 1H)	124.09
14	-	125.98
15	3.98 (s, 2H)	68.33
16	1.72 (s, 3H)	13.37
17	1.36 (s, 6H)	27.80
18	1.36 (s, 6H)	27.80

* The assignments were based on HSQC and HMBC data.

**Table 3 molecules-27-08074-t003:** Cytotoxic activities of isolated compounds from *A. nitrosa* and *A. marschalliana* against the A-549 cell line.

Compounds of *A. nitrosa*	Inhibition against A-549 (%)		Compounds *A. marschallina*	Inhibition against A-549 (%)	
	25 μM	1 μM		20 Μm	2 μM
**2**	23.7	24.8	**2’**	<1	<1
**4**	28.8	30.0	**3’**	<1	<1
**5**	8.7	21.5	**4’**	10.42	<1
**6**	<1	<1	**5’**	3.07	<1
**9**	22.7	24.4	**8’**	<1	<1
**11**	4.5	4.7	**10’**	ND ^a^	ND ^a^
**-**	-	-	**12’**	<1	<1
ADT ^b^		84.4	ADT ^b^		86.1

^a^ ND indicates not determined. ^b^ ADT indicates positive control.

**Table 4 molecules-27-08074-t004:** Cytotoxic activities of isolated compounds from *A. nitrosa* and *A. marschalliana* against the HL-60 cell line.

Compounds of *A. nitrosa*	Inhibition against HL-60 (%)		Compounds *A. marschallina*	Inhibition against HL-60 (%)	
	25 μM	1 μM		20 Μm	2 μM
**5**	8.7	21.5	**4’**	13.04	<1
**6**	18.2	<1	**5’**	<1	<1
**7**	8.7	21.5	**6’**	2.12	<1
**11**	<1	1.9	**10’**	<1	<1
**-**	-	-	**12’**	6.16	<1
ADT ^b^		82.5	ADT ^b^		84.0

^a^ ND indicates not determined. ^b^ ADT indicates positive control.

## Data Availability

Not applicable.

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
