# Peer review of "Secondary Metabolites and Their Cytotoxic Activity of *Artemisia nitrosa* Weber. and *Artemisia marschalliana* Spreng."

_molecules, 2022, doi:10.3390/molecules27228074_

Round 1
Reviewer 1 Report
The authors performed an interesting, but very simple and descriptive study of the phytochemical constitution of aereal parts of two Artemisia species, little studied so far. The authors evaluated also the citotoxity activity of the isolated compounds against cancer cell lines. Despite the novelty of studying barely studied species, the paper needs a lot of corrections and improvements before its acceptance by Molecules. I have concerns regarding the English (several sentences are not clear or are even grammatically wrong), about the lack of some details in M&M section (mainly regarding the analytical methods), as well as lack of a proper discussion (that is not presented at all), what makes the paper merely descriptive. In addition, the compounds did not show citotoxity against cancer cells, which means that they are not potential candidates against this disease. As the authors did not evaluated healthy cells (such as macrophages), it brings few information about other possible utilization of the compounds and consequently of the extracts. Would the extracts be safe to be used in pharmacological preparations? All my specific comments can be found in the PDF here attached. Based on what is mentioned above, I recommend the rejection of the paper. Maybe it could be corrected and submitted as a Communication, instead of as an Article.

Author Response
Thank you so much for your kind letter, regarding our manuscript, together with comments from the reviewers. Our detailed response to the reviewer 1 comments was as follows.
Response to Reviewer 1
→ Thank you for your careful estimation of our manuscript, comments were answered below.
The abstract must be rewritten and organized. It misses an introduction, an aim and should present in a clearer way the results, especially of the cytotoxic tests.
→ Based on these comments, we made the necessary changes to the abstract and clearly defined the aim and introduction.
Not clear. The abstract also needs an introduction, aim and methods. The number of the compounds should not be in the abstract.
→ Corrected the introduction, purpose and methods. The number of connections was removed from the annotation.
Italics.
→Species names of plant species were italicized through manuscript.
These species only occur in this region? Please clarify.
→ In the introduction, the habitat of these species was indicated.
Please clearly include the aim of the study.
→ The aim of the study has been clearly included.
Did the authors also checked the cytotoxicity in health macrophages?
→ Most isolated compounds did not show cytotoxicities or showed insignificant activities, thus experiment on normal cells had not been carried out. Thank you for your suggestions.
Due to the pandemic, the border between China and Kazakhstan is closed. All our extracts and compounds have remained in the State Key Laboratory of Drug Research, Shanghai Institute of Materia Medica, Chinese Academy of Sciences in China. For that reason our researcher and students couldn’t visit China to continue this bioactivity study, China side also couldn’t send back our plant extracts and compounds to Kazakhstan.
The authors must include details of each analytical method used, not only the equipments, but the detailed analytical conditions. This information must be presented in a specific subsection.
→ Details of each analytical method used have been included. The information has been presented in a specific subsection.
How many plants were collected? This 14 kg was a pool of how many plants?
→ The article has been corrected according to the comment. The sentences has been changed.
Why did the authors resuspended the extract in water if they extract in 95% EtOH?
→ The purpose of dissolving in water is to make partition well with the organic solvents. You might think that the extract is insoluble in water, but if we heat the extract slightly in a water bath, all the extract will be dissolved successfuly. The purpose of using water is to create a better partition with organic solvents of different polarities so that the substances of the two parts are well separated.
Check. It should not be 3. It should keep being a subsection of M&M. Besides, did the authors check the citotoxicity in normal cells? It is important to ensure that the compound is not toxic to normal cells.
→ The article has been edited according to the comment.
Please rewrite as a paper, not like a protocol. Besides, include information of positive and negative controls.
→ Has been rewritten as a paper, has been included information of positive control.
Any discussion is presented. The authors must include discussion in their paper. The study is very descriptive.
→ The article has been edited according to the comment.
The authors must cite the supplementary material of the results of their NMR analysis. In addition, they must also provide the data of the other analytical methods.
→ The article has been corrected according to the comment.
This table is not clear at all. Specially the compounds collunm. Besides, the authors must let clarify that the values are in %. In addition, the authors must present the negative control results and a statistical analysis between the results against the control
→ By remark eliminated table 3 -Inhibition rate of A-549 cell lines proliferation of isolated compounds from Artemisia nitrosa and Artemisia marschalliana and table 4 - Inhibition rate of HL-60 cell lines of isolated compounds from Artemisia nitrosa and Artemisia marschalliana.
And this part has been corrected according to the comment.
Therefore, these compounds are not candidates to be anticancer compounds. What about normal cells? Could these compounds be applied in a secure way?
→ Most isolated compounds did not show cytotoxicities or insignificant activity, thus experiment on normal cells had not been carried out. Thank you for your suggestions.
→ We introduced the separation scheme into the article, and corrected the English language.
→In addition, we have provided with 1HNMR and 13CNMR data for known compounds, as well as UV, MS, and full 1D, 2D NMR spectroscopic analyses for new compounds, please see the supplementary information.
Reviewer 2 Report
Through this study, titled “Secondary Metabolites and Their Cytotoxic Activity of Artemisia nitrosa Weber. And Artemsisa marschalliana Spreng.”, a systematic phytochemical investigations of the two Artemisia species were conducted, resulting total two new and twenty-one known compounds. All isolates were evaluated for cytotoxicity against human cancer cell lines 76 HL-60 and the A-549. For readers' understanding and publication in the journal, appropriate answers and supplements to the following are required.
1. References to the methods used for component extraction in this study are lacking.
2. In using each single component in the experiment, it is desirable to present previous study or scientific criteria as a reference for the selection of the concentration (dose).
3. The cytotoxic effect on a specific cancer cell line is also important, but as it is the first extracted ingredient, the safety evaluation part for cells must also be included.
4. Results and presentation of the mechanism of action of all isolates in this study are lacking.
5. Overall English proofreading of the manuscript is required.
Author Response
Thank you so much for your kind letter, regarding our manuscript, together with comments from the reviewers. Our detailed response to the reviewer 2 comments was as follows.
Response to Reviewer 2
Through this study, titled “Secondary Metabolites and Their Cytotoxic Activity of Artemisia nitrosa Weber. And Artemisa marschalliana Spreng.”, a systematic phytochemical investigations of the two Artemisia species were conducted, resulting total two new and twenty-one known compounds. All isolates were evaluated for cytotoxicity against human cancer cell lines 76 HL-60 and the A-549. For readers' understanding and publication in the journal, appropriate answers and supplements to the following are required.
→ Thank you for your careful estimation of our manuscript, comments were answered below.
References to the methods used for component extraction in this study are lacking.
→ In this revision manuscript, the numbers of references were increased to 48. The references have been added to the Introduction, Materials and Methods sections.
In using each single component in the experiment, it is desirable to present previous study or scientific criteria as a reference for the selection of the concentration (dose).
→ This biological activites study was conducted at the Shanghai Institute of Materia Medica, China. Employees of this institute did not present previous studies when using each individual component in the experiment.
The cytotoxic effect on a specific cancer cell line is also important, but as it is the first extracted ingredient, the safety evaluation part for cells must also be included.
Results and presentation of the mechanism of action of all isolates in this study are lacking.
→ This biological activites study for isolated compounds was conducted at the Shanghai Institute of Materia Medica, China.
Due to the pandemic, the border between China and Kazakhstan is closed. All our extracts and compounds have remained in the State Key Laboratory of Drug Research, Shanghai Institute of Materia Medica, Chinese Academy of Sciences in China. For that reason our researcher and students couldn’t visit China to continue this bioactivity study, China side also couldn’t send back our plant extracts and compounds to Kazakhstan.
Overall English proofreading of the manuscript is required.
→ Taking into account your comments about the English language, we worked out and eliminated grammatical errors.
→ We have added one more coauthor, and the list of co-authors has been corrected.
Round 2
Reviewer 1 Report
The authors answered and corrected all my suggestions. Some other comments can be found in the PDF file. A proper discussion of the results keeps missing and it must be included.

Author Response
Thank you so much for your kind letter dated Oct 27, 2022, regarding our manuscript, together with comments from the reviewers. Corrections were colored in red. Our detailed response to the reviewer’s comments was as follows.
Response to Reviewer 1
→ Thank you for your careful estimation of our manuscript, comments were answered below.
Italics and size of a front.
→During the conversion of word document to a pdf format some words or phrases may also undergo automatically changes, including italics and size. Please refer to the word document to prevent such inconsistencies.
Compared to what? If the authors are using superlative they must clarify or rephrase the sentence.
→The sentence had been rephrased
The authors must include the details of the LC-MS analysis: column, volume of injection, temperatures, energy…
→The details have been added
With flowers? Or in vegetative stage? The authors could present pictures of both species, since they are poorly studied in literature
→. The whole plant had been investigated, include aerial and underground parts of one. Pictures of the two artemisia species were included in Supplementary information.
Once again the discussion is not presented. The authors must include some discussion about their results. Were these compounds already described in other species of Artemisia or other Asteraceae species? Have these compounds some important biological activity that could be related to some popular uses? Or that could be attribute as bioactive compound? For the cytotoxic activity, if the compounds do not present this activity, which other compounds (not evaluated) could be proposed as the cytotoxic in these Artemisia species? If even the extracts are not cytotoxic at all (according to the literature), how it could be important for use of the extracts from people? Could these extracts be applied as anticancer agents even without this property? The paper needs discussion.
→ The discussion had been revised according to the suggestions.
Include in the discussion if the compounds were already isolated in other Artemisia species or if it has biological activities.
→ The discussion has been edited according to the comment.
Do both species occur at the same habitats or regions?
→ The information had been added.
Reviewer 2 Report
It is unfortunate that additional experiments did not proceed smoothly due to the pandemic, but there are experiments that need to be added to be published in this journal.
Author Response
Mr. Hyland Xue. Assistant Editor, MDPI Nanjing
Dear Mr. Hyland Xue.
Thank you so much for your kind letter dated Oct 27, 2022, regarding our manuscript, together with comments from the reviewers. Corrections were colored in red. Our detailed response to the reviewer’s comments was as follows.
Response to Reviewer 2
It is unfortunate that additional experiments did not proceed smoothly due to the pandemic, but there are experiments that need to be added to be published in this journal.
→ The compounds had been evaluated for anti-inflammatory activity and results added to the manuscript and Supplementary information.